# Evaluation of the Vibration Signal during Milling Vertical Thin-Walled Structures from Aerospace Materials

**DOI:** 10.3390/s23146398

**Published:** 2023-07-14

**Authors:** Szymon Kurpiel, Krzysztof Zagórski, Jacek Cieślik, Krzysztof Skrzypkowski, Witold Brostow

**Affiliations:** 1Faculty of Mechanical Engineering and Robotics, AGH University of Science and Technology, Mickiewicza 30 Av., 30-059 Krakow, Poland; 2Faculty of Civil Engineering and Resource Management, AGH University of Science and Technology, Mickiewicza 30 Av., 30-059 Krakow, Poland; 3Laboratory of Advanced Polymers & Optimized Materials (LAPOM), Department of Materials Science and Engineering, University of North Texas, 3940 North Elm Street, Denton, TX 76207, USA; 4Department of Physics, University of North Texas, 3940 North Elm Street, Denton, TX 76207, USA

**Keywords:** acceleration vibration, milling, vertical thin-walled elements, short-time Fourier transform, aerospace material, titanium alloy, nickel alloy

## Abstract

The main functions of thin-walled structures—widely used in several industries—are to reduce the weight of the finished product and to increase the rigidity of the structure. A popular method for machining such components, often with complex shapes, is using milling. However, milling involves undesirable phenomena. One of them is the occurrence of vibrations caused by the operation of moving parts. Vibrations strongly affect surface quality and also have a significant impact on tool wear. Cutting parameters, machining strategies and tools used in milling constitute some of the factors that influence the occurrence of vibrations. An additional difficulty in milling thin-walled structures is the reduced rigidity of the workpiece—which also affects vibration during machining. We have compared the vibration signal for different approaches to machining thin-walled components with vertical walls made of Ti6Al4V titanium alloy and Inconel 625 nickel alloy. A general-purpose cutting tool for machining any type of material was used along with tools for high-performance machining and high-speed machining adapted for titanium and nickel alloys. A comparison of results was made for a constant material removal rate. The Short-Time Fourier Transform (STFT) method provided the acceleration vibration spectrograms for individual samples.

## 1. Introduction

There is a tendency to lower the weight of structures—hence a noticeable interest in using thin-walled components. Such structures are usually made of materials such as aluminum and titanium alloys, although some advanced materials also are used such as nickel alloys or else some grades of stainless steel [1]. Based on forecasts, it is expected that interest in thin-wall structures will continue at a high level in the coming years. This is due to the lack of alternative materials for high-temperature components such as turbines or aerostructure engines [2]. Various definitions of thin-walled components are available—involving the dimensions of the semi-finished product. For cylindrical elements, there is a definition saying that if the ratio of wall thickness to the diameter of the semi-finished product is less than 1/10 then the element is defined as thin-walled. For plates, it is assumed that if the ratio between the length of the shorter side and the wall thickness is in the range of between 1/100 and 1/5, then one has a thin-walled element [3,4]. 

One of the main types of machining used to manufacture such parts—often of complex shape—is milling. There are many undesirable phenomena, and one of them is vibration, which has a negative effect on the quality of the machined surface. Vibrations are caused by moving parts. During milling, each element affects the rigidity of the system machine–tool holder–workpiece–tool [5]. The vibrations strongly affect the surface roughness and have a significant impact on tool life [6]. Cutting parameters, machining strategies and the cutting tools used are just a few of the factors that have an impact on vibration [7]. All machine components have natural frequencies, i.e., frequencies at which the system vibrates when brought out of equilibrium [8]. A typical classification of vibrations distinguishes free, forced and self-excited vibrations, which are caused by oscillation of friction values and cutting forces; there are also forces transferring energy from the drive to the machine system [8,9]. In the milling process, the most common type of vibrations are self-excited chatter vibrations [10]—characterized by the fact that they arise suddenly and reach full scale in unstable system [11]. Chatter vibrations during the milling process are manifested by the occurrence of characteristic notching, i.e., characteristic surface irregularities. Their formation can be caused by the instability of both the workpiece and the tool [12,13,14,15,16,17,18]. The effect of chatter vibration is the reduction of machining productivity and dimensional accuracy, deterioration of the quality of the machined surface, reduction of tool life and production of excessive spindle noise [12,13,14]. Identifying the stability of machining processes can be accomplished using a variety of analysis methods and calculations for measuring vibration during machining involving acceleration sensor, dynamometers and microphones [18,19,20,21].

In engineering research, most of the signal variables are in analog form, while modern measurement equipment allows digital representation of the signal. The digital representation of an analog signal often affects the characteristics of the signal; hence, it is necessary to understand the basic principles of signal processing [17]. A key tool in signal analysis, as well as in the control of the processing process, is FFT (Fast Fourier Transform) analysis [18,22]. FFT is unable to represent signal changes over time and only shows an averaged value—the main problem with using this tool [23]. For this reason, the Short-Time Fourier-Transform (STFT) can be used as an alternative tool to present a full spectrum of changes during processing. The principle of the STFT is based on the Discrete Fourier Transform (DFT), which represents the frequency and phase components of a time-dependent signal segment. A discrete STFT can be obtained by using equal segmentation with overlap windowing techniques and performing a DFT [24].

Using spectrogram graphs to study conditions during processing is becoming more common—although this tool is not as widely used as the Fast Fourier Transform. Wan and coworkers [25] presented time–frequency spectrograms of the resultant force for parameters in the stable and unstable operating range during the milling of carbon fiber-reinforced plastic. Li and coworkers [26] described the possibility of using spectrograms during vibration control in the grinding process and correlated the results so obtained with micro-surface topography. The Short-Time Fourier-Transform can also be used to control the turning process. Yan and coworkers [27] provided an approach based on using the spectrogram to evaluate the vibration frequency of a workpiece made of reinforced steel during the turning process and correlated the results with the roughness parameters obtained. During the machining of thin-walled workpieces, vibrations may appear irregularly or may be increased due to the reduced stiffness of the workpiece [1].

We inferred from the publications cited above that it might be worthwhile to use STFT in the milling process of thin-walled structures. STFT will allow providing information on the course of vibrations along the entire length of the specimen.

The main purpose of our study is the determination of the vibration distribution and evaluation of the stability of the milling process of parts with vertical thin walls—using tools designed for various machining methods. We obtain spectrograms of the sample acceleration’s vibration signal in the coordinate system of component axes during the milling of thin-walled structures. The input variables are the type of material used (titanium or nickel alloys), the cutting strategy (larger involvement of the face part or larger involvement of the cylindrical part) and the type of cutting tool (general purpose, performance machining and high-speed machining).

## 2. Materials and Methods

The results presented in this paper are part of a broader study. The input factors used here were the workpiece material, the machining strategies and the cutting tools used to machine vertical thin-walled parts. On this basis, we have milled samples with thin vertical walls. Appropriate vibration tests were carried out for samples so prepared. From the obtained waveforms, we have created vibration spectrograms of the sample’s acceleration along the three component axes. A diagram presenting the test object along with the input and output factors is shown in Figure 1.

A 10 mm-thick sheet of metal was used for the samples, first ground to a dimension of 9 mm on both sides using a SGA3063AHD surface grinder manufactured in Poland by the company Jazon. Then, semi-finished samples with dimensions of 31 × 51 mm were cut using a WaterJet HWE-P 1520 machine tool supplied by the German company H. G. Ridder Automatisierungs GmbH. In the last step, the overall dimension was carried out on a DMC 1150 V machine (manufactured and provided by German company DMG Mori), resulting in cubes with dimensions of 30 × 50 mm. In the next stage, samples were milled with thin vertical walls—according to the methodology described below, see Figure 2. A vertical wall with a thickness of 1 mm was made along the entire length of the specimen equal to 50 mm and a height equal to 16 mm. The shapes of the semi-finished product and the finished sample dimensions are constant factors in our study.

Another parameter that provides pertinent information on the samples is the material removal rate (MMR) called below Q, which provides the amount of material removed from the workpiece relative to the amount of initial material. It is defined in terms of the feed rate V_f_ (mm/rev), the cut depth a_p_ (mm) and the radial depth a_e_ (mm), [28] according to the following relation Equation (1): Q = V_f_ × a_p_ × a_e_ [cm^3^/min](1)

In our experiments—regardless of the machining strategy—we have maintained Q = 2.03 cm^3^⁄min.

From the group of titanium alloys, we have used Ti6Al4V grade 5, an alloy quite often used [29,30], applicable both in aircraft structures and in engines [31]. This alloy in its annealed state is suitable for applications where the temperature does not exceed 400 °C. It is highly machinable, provides moldability and can be welded by conventional processes [32]. From nickel superalloys, we have chosen Inconel 625. It is a nickel–chromium–molybdenum alloy with the addition of niobium. Molybdenum stiffens the alloy substrate, resulting in high strength without the need for hardening by heat treatment. Inconel 625 is resistant to corrosive environments, including pitting and crevice corrosion [33]. The chemical compositions and mechanical properties of our materials are provided in Table 1, Table 2, Table 3 and Table 4.

We have used several types of cutting tools. Machining was carried out with three 10 mm diameter monolithic milling tools designed for different machining methods—a general purpose tool (for a wide range of materials), as well as a performance tool and a speed tool, which are designed only for machining titanium and nickel alloys [34,35,36]. Table 5 shows the tools adopted for the experiment. The geometry specifications of each tool are shown in Table 6.

Significant in our work was the cutting strategy. Two different approaches were used to utilize the capabilities of the cutting tool. In the first approach, it was decided to engage the face part of the tool to a larger extent by using a depth of cut of 4 mm and a radial depth of 2 mm. The second approach involved the cylindrical part of the tool to a larger extent using a depth of cut equal to 16 mm and a radial depth of 0.5 mm. Table 7 shows the cutting parameters adopted for each of the samples made. The adopted feed rate V_f_ was constant for all cases and equaled 255 mm/min while the cutting speed V_c_ was in the middle of the speed range provided for each of the cutting tools used by their manufacturer. Thus, the cutting speed equaled 100 m/min for the titanium alloy and 40 m/min for the nickel alloy. Due to the higher stiffness of the nickel alloy, lower cutting speeds had to be used. The parameters adopted during the experiment were within the range recommended for use during machining.

SILUB MAX two-component water–oil emulsion was used—following the cutting tool manufacturer’s recommendations. The coolant meets the requirements of TRGS 611 and is designed for general applications including milling of materials such as titanium and nickel alloy under extreme conditions [37]. A mixture of 15 vol.% oil and 85 vol.% water was used when machining the samples.

The samples were made on a Mikron VCE 600 Pro numerically controlled milling machine manufactured by GF Machining Solutions (Switzerland), supplied with iTNC530 control software developed by Heideinhain (Germany). The prepared semi-finished product, in the form of a cube measuring 9 mm × 30 mm × 50 mm, was mounted in a vise at 10 mm of the specimen height. During the execution of samples with thin vertical walls, the vibration sensor was mounted on a vise. The tools were mounted in an ER32 tool holder using a ϕ = 10 mm precision collet, and the tool overhang was constant for all tools. The experimental setup and method of mounting the sensor are shown in Figure 3.

The vibration signal occurring during machining was recorded using a circuit consisting of a PCB-356B08 vibration sensor manufactured by PCB Piezotronics Inc. (USA) connected to an NI USB-9162 measurement card. The card transmitted the signal to a computer. The signal was recorded in a measurement program prepared in the LabView environment. The system for measuring vibrations during sample processing is shown in Figure 4.

The vibration sensor was calibrated according to the calibration data sheets provided by the manufacturer. The adopted values and data for the sensor are shown in Table 8.

During the processing of the samples, the acceleration on each of the three axes of the coordinate system was recorded at a sampling rate of 25,000 Hz. In the first step, the original vibration waveforms were determined—which made it possible to follow the acceleration during each of the transitions along all the component axes of the *x*, *y*, and *z* coordinate systems. Then, the analysis of the results was performed using the Short-Time Fourier-Transform (STFT) program prepared in MatLAB R2020B (update 3—9.9.0.1538559) software. To evaluate the phenomena occurring during machining in specific areas of the machined surface, it is advantageous to correlate them with the conditions during machining. STFT provides an opportunity to achieve this goal since spectrograms with changes in amplitudes over time are visible. In the case of the classical Fourier analysis (DFT and FFT), only the average value for the entire process is obtained. It is for this reason that it was decided to use spectrograms in our study.

As part of the analysis of the results, the spectrograms of the sample acceleration vibration signal were determined to evaluate the conditions during sample machining for the entire frequency range for the last pass—the finishing pass—which determines the surface of the finished product. The analysis of the last pass was presented for both the input side and the output side. To perform spectrograms, the built-in spectrogram function in MatLAB with the Hanning window was used. This window was selected because of its popularity in vibration tests during the milling process, which was apparently because of its good performance. This window is also a good choice because it features a falling profile that minimizes the effect of spectrum leakage. Spectrum leakage can cause unwanted artifacts and distortion of the frequency spectrum. Moreover, the Hanning window provides a good balance between time and frequency resolution. The frequency range of 0.5–5000 Hz was used to implement the window, in line with the frequency range of the vibration sensor. To perform this function, the signal from the entire transition was used, and the subsequent parameters depended on the “framelen” variable equal to 471 for titanium alloy and 295 for nickel alloys. These values were adopted since they are the number of samples per full period of vibration resulting from the spindle’s rotary motion. It was assigned as a variable value because subsequent parameters were dependent on it. Equation (2) below shows the line from MatLAB taken to prepare the spectrogram. Here “Data” is the input signal from the measurement; hann(4∙framelen) defines the window used and shows its length during the analysis; 2∙framelen is the offset between the consecutive segments of the signal (a value of 2∙framelen means that the segments will have 50% overlap); 4∙framelen is the length of the segment of the signal that will be subjected to the Fourier transform while fs is the data signal sampling frequency. We have: [Sw, W] = spectrogram(Data, hann(4∙framelen), 2∙framelen, 4∙framelen, fs)(2)

In the above equation, Sw is defined as a matrix, which stores short-time spectra for individual segments of the signal, and W is a vector containing the values of time midpoints for each segment.

As can be seen in Equation (2), four times “framelen” was taken for the Hanning window, while the overlap was twice the value of framelen and half of the length of the window. Signals from transitions that had a significant effect on the surface of the finished product were included in the analyses; hence, the last full transition for the input side and the output side were considered. 

The classical STFT shows a plot of frequency amplitudes versus measurement time. Within the framework of this study, this value was related to the sample length S in terms of the feed rate V_f_ and time t Equation (3):S = V_f_ × t [mm](3)

During the experiment, a constant feed rate value V_f_ = 255 mm/min was maintained. The time t in Equation (3) was calculated in terms of the length of the measurement signal L and the sampling frequency f_s_ adopted during the measurement, namely:t = (linspace(0, L − 1, L))/f_s_ [s](4)

## 3. Results and Discussion

The specimens so obtained with thin vertical walls show certain characteristics on the machined surfaces. A chamfer appeared at the point of leaving the tool from the material; this occurred on both sides of the workpiece, that is on the entry and exit sides of the tool from the material. A photo of the aforementioned chamfer is shown in Figure 5.

### 3.1. Vibration Spectrograms for Samples from Titanium Alloy

We consider first the results obtained for the titanium alloy. Spectrograms for such vertical thin-walled samples for a milling strategy with larger involvement of the face part are shown in Figure 6, Figure 7 and Figure 8 and with larger involvement of the cylindrical part in Figure 9, Figure 10 and Figure 11.

Focusing on the spectrogram for specimen T1_1 (Figure 6) machined using a general-purpose tool with a milling strategy with larger involvement of the front part, vertical striations on both sides are noticeable. This indicates the appearance of chatter vibration. For this sample, higher irregularity is observed on all three axes on the output side, while a marked increase in amplitudes appears on the input side on the *x*-axis for sample lengths of 10–12 mm. In regular processing, there should occur horizontal striations, associated with amplitudes in the corresponding frequency band. A similar geometry to the dedicated general-purpose tool is characterized by the high-performance machining tool used to mill sample T2_1 (Figure 7). In the case of this sample, a relatively regular pattern of frequency bands is seen, which is similar in nature between the input and output sides. However, we see a vertical blurring of the frequency bands along the length of the sample of about 15 mm, which may indicate the presence of chatter vibrations. Sample T3_1 (Figure 8) milled with the high-speed machining tool using the same strategy as the two previously mentioned samples gave the least favorable result due to the wide spectrum of frequency bands present with a highly irregular distribution. The occurring acceleration amplitudes take on high values, and their distribution itself shows the occurrence of harmonic oscillations. This indicates that apparently, vibrations of the thin-walled structure occurred during the processing of this surface.

The surfaces of the samples machined by side milling with larger involvement of the front part, i.e., samples T1_1–T3_1, have features revealing the occurrence of chatter vibrations—confirmed by the spectrogram graphs. For the second strategy, an effect of these vibrations is not observed.

More favorable results for titanium alloy—in terms of minimizing vibration during machining—were observed for strategies with larger involvement of the cylindrical part. Samples T4_1 and T5_1 (Figure 9 and Figure 10) made with a general-purpose tool and a high-performance machining tool, respectively, show stable frequency bands, and their distribution is similar both between the two samples and the machined sides (input and output sides). However, analyzing in more depth, it is seen that slightly higher values of acceleration amplitudes occur for the input side for both samples. The assumed values of amplitudes for samples machined with the side milling strategy with larger involvement of the cylindrical part T4_1–T5_1 seem relatively low compared to the opposite strategy. The sample made with the high-speed machining tool (T6_1—see Figure 11) presented an unfavorable result. Although the bands adopt a more regular distribution on the length of the sample than for sample T3_1 (Figure 8), a wide spectrum of frequencies present is observed, indicating instability in the process. The distribution of the bands for sample T6_1 is irregular but with a more stabilized distribution than for T3_1.

During the visual evaluation of samples T4_1–T6_1 (Figure 9 and Figure 11) obtained during milling with greater involvement of the cylindrical part, the surface characteristic of chatter vibration was not observed. Therefore, it is concluded that the strategy has a significant effect on this phenomenon.

In the spectrogram graphs, the largest amplitudes are seen on the *x*-axis, i.e., in the direction normal to the thin wall being machined. This is an expected effect, due to the phenomenon of wall deflection. The smallest amplitudes are seen on the *z*-axis and a large number of amplitudes are also seen on the *y*-axis (on the feed direction).

The graphs show the areas of the tool’s penetration into the material and its exit from the material. We see irregularities in amplitudes near the ends of the graph. This is due to the variable radial width of the cutter engaged during surface machining. This signal is related to the occurrence of chamfer at the end of the sample (Figure 5).

### 3.2. Vibration Spectrograms for Samples from Nickel Alloy

Similar to titanium alloy, spectrograms were determined for nickel alloy samples with vertical thin-walls machined with larger involvement of the face part and cylindrical part, respectively; see Figure 12, Figure 13, Figure 14, Figure 15, Figure 16 and Figure 17.

Samples N1_1 and N2_1, milled with a general-purpose tool and a high-performance machining tool, respectively, using a side milling strategy with larger involvement of the face part (Figure 12 and Figure 13), are characterized by the occurrence of amplitude bands in a similar frequency range. From the presented spectrograms, we see that sample N2_1 exhibits lower values of acceleration amplitudes. For both samples, vertical fuzzy bands are observed on both sides of the treated *x* and *y* axes. In this milling strategy, the highest values of amplitudes are visible for sample N3_1 made with the high-speed machining tool (Figure 14), for which both fuzzy frequency bands and vertical striations are observed—indicating the instability of the machining process under such cutting conditions.

For all samples machined by the side milling strategy with larger involvement of the face part, ignoring the presence of vertical fuzzy striations, a similar distribution of frequency bands is observed between the input and output sides. The striations are more pronounced on the output side, which may indicate that the sample begins to vibrate in the next pass.

A similar mutual distribution between the machined sides is also observed for samples processed with the second strategy—side milling with larger involvement of the cylindrical part, i.e., for samples N4_1–N6_1 (Figure 15, Figure 16 and Figure 17). Between these samples, we see similar locations of the frequency bands. This situation was not observed for titanium alloy samples, where specimens made with the high-speed machining tool were characterized by significantly higher deviations compared to those machined with other tools. On this basis, it is concluded that the stiffness of the material has a significant effect on machining vibration. For specimens milled by side milling strategy with larger involvement of the cylindrical part, a regular and stable machining process is observed.

We infer that the qualitatively most favorable machining conditions were for samples N4_1 and N5_1 (Figure 16 and Figure 17). The largest amplitudes with a wider spectrum are observed in both strategies for the tool dedicated to high-speed machining (N3_1 and N6_1 in Figure 14 and Figure 17), but their deviations from other tools are not as significant as for titanium alloy samples.

Based on the above information, we infer that nickel alloy specimens show a higher regularity of frequency bands compared to those for titanium alloy—a result of the comparable mutual distribution of frequencies and their amplitudes on both ma-chined sides. Apparently in the case of nickel alloy—which is a material of considerable stiffness—the machined side has hardly any influence on the course of vibrations. Some signal analogies can be seen between tools for opposite machining strategies; i.e., their distribution is very similar, with higher amplitudes for the milling strategy with larger involvement of the face part (Figure 12, Figure 13 and Figure 14) than the cylindrical part (Figure 15, Figure 16 and Figure 17).

At the beginning and end of the length of the samples for nickel alloy, decreases in frequency amplitudes on the vibration spectrograms are visible, but their effect is lower than for titanium alloy samples. On the finished product, a chamfer was noticed at these locations—a result of the tool entering and leaving the material (Figure 5).

For the nickel alloy samples, chatter vibrations are seen on the surfaces of samples N4_1 (Figure 15) and N6_1 (Figure 17), machined using the strategy with larger involvement of the cylindrical part—with the tool dedicated to general purpose and tool for high-speed machining, respectively. It is interesting that no chatter vibration effect is seen on the surface of the nickel alloy samples for which the tool for high-performance machining was used (Figure 16).

### 3.3. Statistical Analysis of Results

Based on the data and the results presented in Section 3.1 and Section 3.2, it is difficult to clearly indicate which processing parameters for the vertical thin wall samples provide the most favorable results. Therefore, it was decided to conduct a statistical analysis of the results. As part of this analysis, the maximum values of the three components of the vibration amplitudes of acceleration were taken for all full passes made in the machining of the thin-walled specimens (16 passes). First of all, the graphs of average values including the standard error and the standard deviation were obtained, shown in Figure 18. In parallel, the basic statistical parameters for each case were determined—shown in Table 9, Table 10 and Table 11.

The statistical results obtained, including those pertaining to individual axes, allow us to reach several conclusions:On the *x*-axis: for each milling and tool type (GP, PM, and SM), the mean value is close to 50; the median for most cases is close to the mean, suggesting a symmetrical distribution of the data. The minimum and maximum values vary considerably between different milling types and tools. The variance and standard deviation are relatively large compared to the mean values, indicating a relatively wide dispersion in the data.On the *y*-axis: the average values oscillate in the range of 30–140, depending on the type of milling and tool. The median is close to the mean for most cases. The minimum and maximum values vary considerably between different milling types and tools. The variance and standard deviation are relatively large, indicating again a dispersion of the data.On the *z*-axis: the average values oscillate in the range of 40–180, depending on the type of milling and tool. The median is close to the mean for most cases. The minimum and maximum values vary considerably between different milling types and tools. The variance and standard deviation are also relatively large, indicating the dispersion of the data.

Focusing on input factors, we conclude:The largest number of deviations and errors are observed for samples milled using a high-speed machining tool.Generally, the smallest scatter of values is observed for samples milled using a high-performance tool. Based on the graphs presented, it is recommended to use this tool during milling under such conditions.Smaller standard error and standard deviation are obtained for samples made using a strategy with greater involvement of the cylindrical part.For the nickel alloy, lower values of errors and deviations are observed on the *x*-axis—compared to the titanium alloy. For acceleration amplitudes on the other axes (*y*, *z*), these values are higher for titanium alloy samples.

Based on obtained results from statistical analysis, the average values of the amplitude of acceleration on each axis for all samples were prepared and presented in Figure 19.

Given the average values of vibration frequency amplitudes presented in Figure 18, the lowest results for both materials on all three axes are seen for a tool designed for high-performance machining using a side milling strategy with larger involvement of the cylindrical part. Lower values were obtained for nickel alloy samples—a consequence of the higher rigidity of the workpiece material. The average values of the amplitude of acceleration for the nickel alloy samples—compared to the titanium alloy—were lower on the *x*-axis by about 20%, on the *y*-axis by about 5%, and on the *z*-axis by about 60%. When planning the machining process for minimizing vibrations, it is most advantageous to reach for the mentioned tool and machining strategies. The highest values of average vibration amplitudes were obtained for the high-speed machining tool during milling with the assumed parameters and conditions. This shows that increasing the cross-sectional area of the cut layer with the use of a tool of such geometry has a negative impact on the machining conditions, which can have an adverse effect on the surface of the finished product.

## 4. Summary and Concluding Remarks

According to the plan, samples with vertical thin walls were milled under controlled machining conditions. During the milling of the samples, the acceleration vibrations of samples were measured on three axes of the coordinate system. The input factors were the material (titanium alloy or nickel alloy), the cutting tool (for general purpose, for high-performance machining and for high-speed machining) and the cutting strategy (side milling with larger involvement of the face part and with larger involvement of the cylindrical part). Based on the measurements performed, spectrograms of the unfiltered signal were prepared and are shown in relation to the length of the sample. The following conclusions seem worth noting.

Comparing the graphs obtained for the first input factor—the workpiece material—the process is more stable for the nickel alloy than for the titanium alloy samples. We have already discussed this above in terms of rigidity and hardness. For nickel alloy workpieces, the distribution of the acceleration frequency amplitudes on the input and output sides are very close to each other. For the assumed cutting conditions, instability of the process is observed during the milling of titanium alloy samples, seen as an irregular course of the amplitudes. The titanium alloy samples present the trend observed for the nickel alloy samples; on the input and output sides, there are similar values of the amplitudes of the acceleration vibration frequencies and their distribution along the length.

Focusing on the second input factor, namely the individual cutting tools and the spectrogram plots obtained for them, the lowest values of acceleration vibration frequency amplitudes are observed for the tool designed for high-performance machining, while the highest values are observed for the tool designed for high-speed machining. The geometry of tools designed for high-performance machining affects the stiffness of the tool and, consequently, the stability of machining. Future studies should focus on determining the interrelation between the stiffness of the tools used and the vibrations recorded.

The last variable factor was the cutting strategy. For both materials, samples made with the strategy with larger involvement of the cylindrical part (labeled 4 to 6) show higher stability and lower amplitude values compared to samples obtained by milling with greater involvement of the front part (labeled 1 to 3). Side milling with larger face part involvement shows a more irregular distribution of vibration amplitudes.

In the future, we would like to look for a correlation between the vibration plots obtained during machining with the cutting forces, as well as the surface topography and dimensional-shape accuracy of the finished product. Thus, our future plans include:Experimenting with a larger number of samples;Carrying out tests with different variances of cutting parameters;Application of damping elements;Checking different approaches to clamping the sample.

While metals have been used for thousands of years [38], new original applications are reported in the literature. Thus, Pereira, Braga and Kubrusly [39] describe an ultrasonic system capable of simultaneously power transferring and transmitting data through a set of two flat steel plates separated by a fluid layer—using a pair of co-axially aligned piezoelectric transducers on opposite sides of the barrier. Ioana and coworkers in Bucharest [40] used physicochemical systems connected to a computer to optimize the melting of metals. Vaskeliene and her colleagues in Kaunas [41] developed high-temperature ultrasonic transducers for non-destructive testing; the transducers are resistant to multiple heating-cooling cycles. Torokhtil and his colleagues in Rome [42] developed a dielectric-loaded resonator for the determination of the root-mean-square surface roughness of metals. Alloy weld solidifications have been discussed by Messler Jr. and De Fazio [43].

To take a broader view, let us consider teaching metals and their alloys as a part of teaching materials science and engineering and also teaching related subjects such as physics, mechanical engineering and chemistry. Recrystallization in metals and alloys has been explained using a simple analytical model by Monge and Worner [44]. The corrosion of metals was discussed by Iribarren Laco and Iribarren Mateos [45]. Titanium we have worked with is used in shape memory alloys—as explained by Wadood and Azair [46].

## Figures and Tables

**Figure 1 sensors-23-06398-f001:**
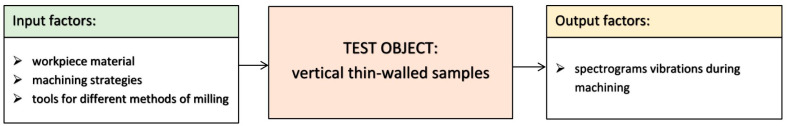
The object model used in the conducted experiment.

**Figure 2 sensors-23-06398-f002:**
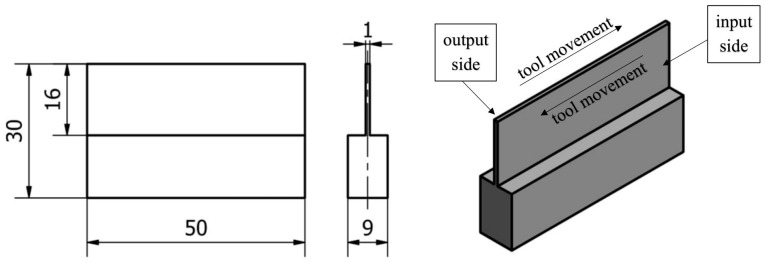
Documentation and description of the vertical thin-walled sample.

**Figure 3 sensors-23-06398-f003:**
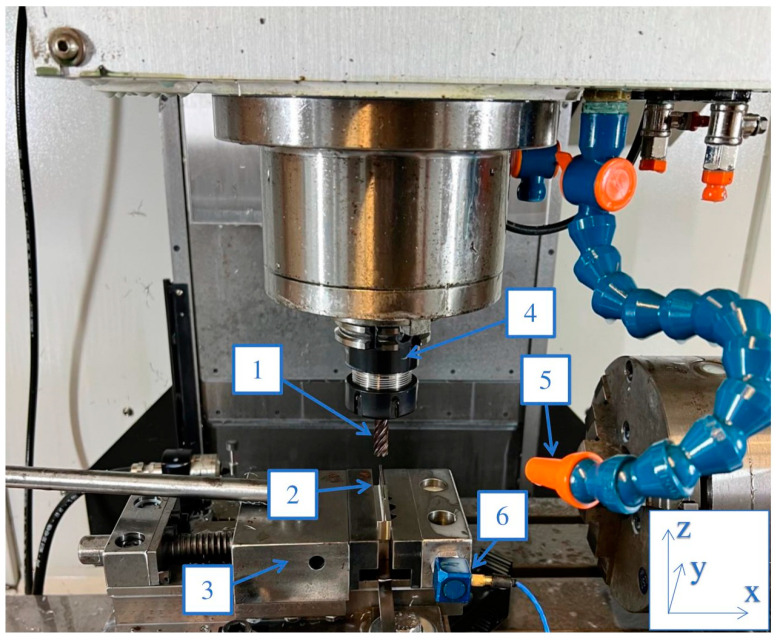
Experimental setup: 1—tool, 2—machined workpiece, 3—vice, 4—tool holder, 5—coolant, 6—piezotronics accelerometer.

**Figure 4 sensors-23-06398-f004:**
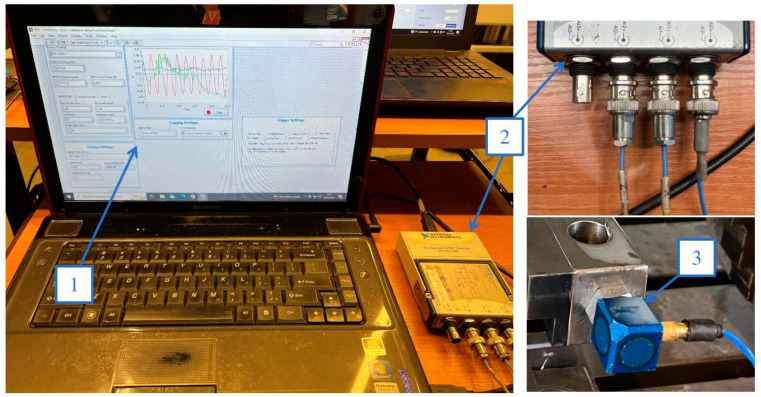
The vibration measurement system used in the experiment: 1—a computer with measurement program in LabVIEW, 2—measurement card NI USB-9162, 3—piezotronics accelerometer PCB-356B08.

**Figure 5 sensors-23-06398-f005:**
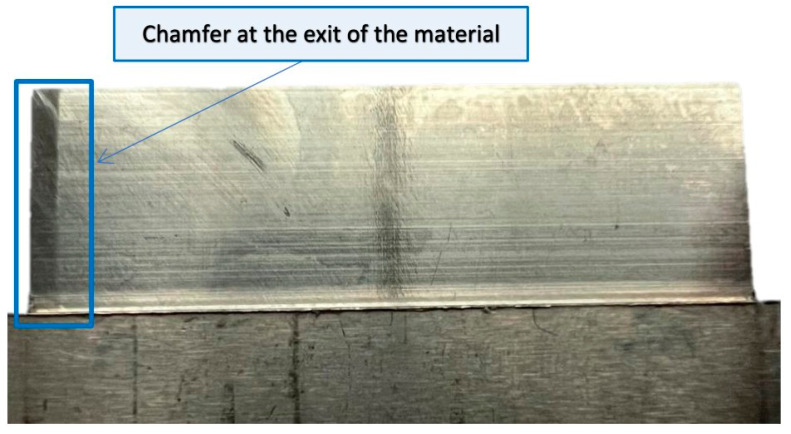
Chamfer at the exit of material for vertical thin-wall sample.

**Figure 6 sensors-23-06398-f006:**
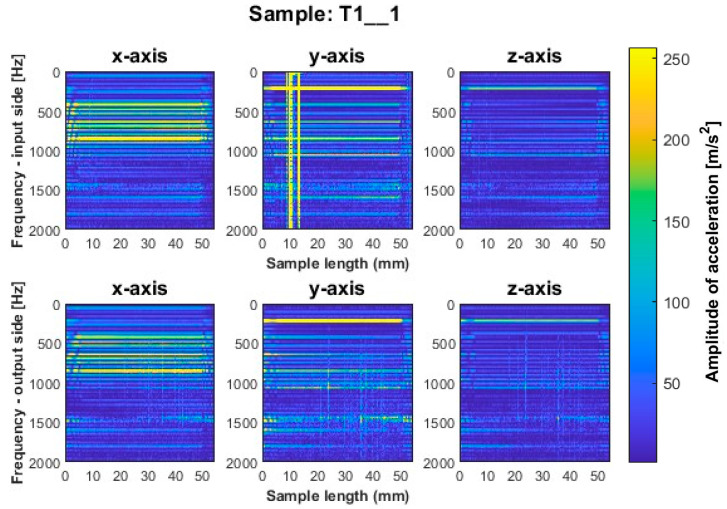
Vibration spectrogram of the acceleration signal for sample T1_1.

**Figure 7 sensors-23-06398-f007:**
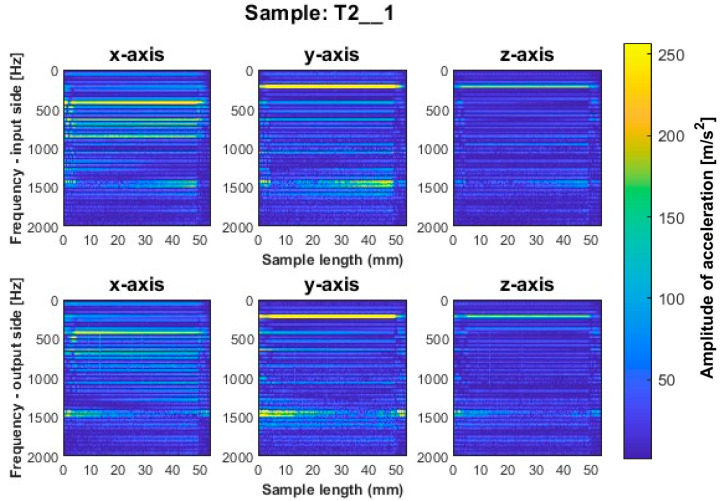
Vibration spectrogram of the acceleration signal for sample T2_1.

**Figure 8 sensors-23-06398-f008:**
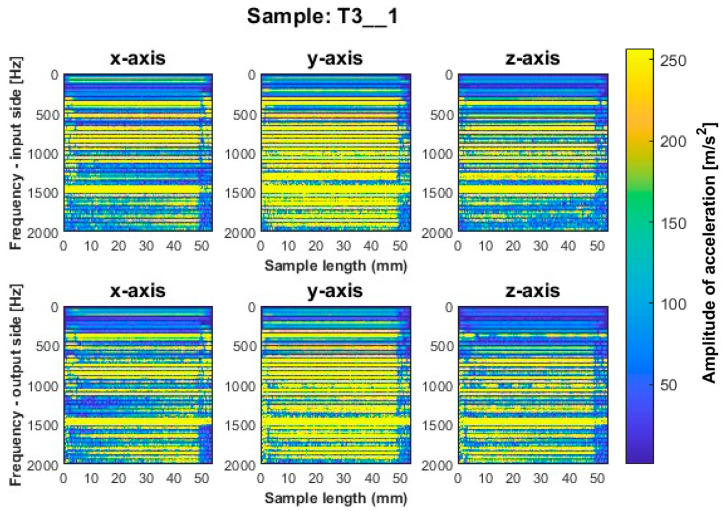
Vibration spectrogram of the acceleration signal for sample T3_1.

**Figure 9 sensors-23-06398-f009:**
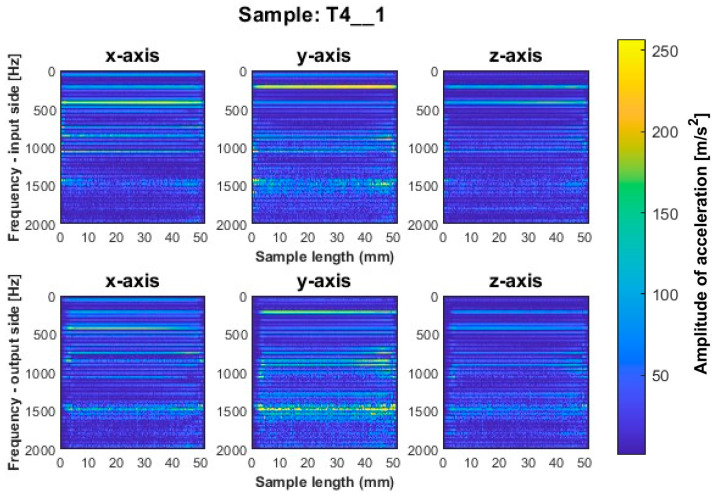
Vibration spectrogram of the acceleration signal for sample T4_1.

**Figure 10 sensors-23-06398-f010:**
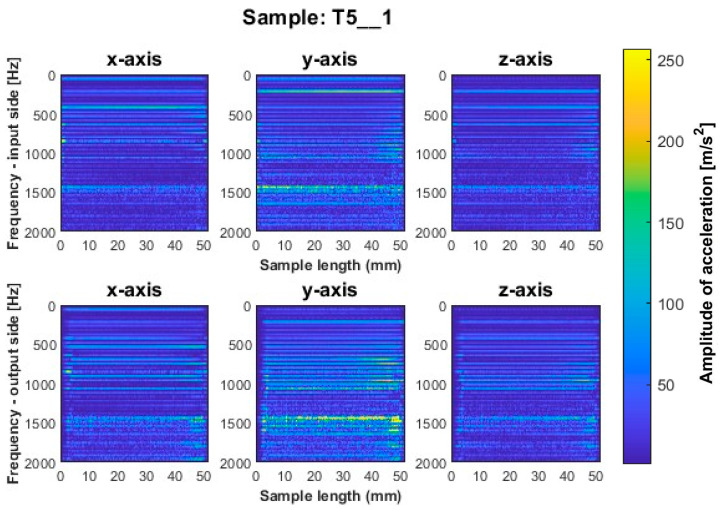
Vibration spectrogram of the acceleration signal for sample T5_1.

**Figure 11 sensors-23-06398-f011:**
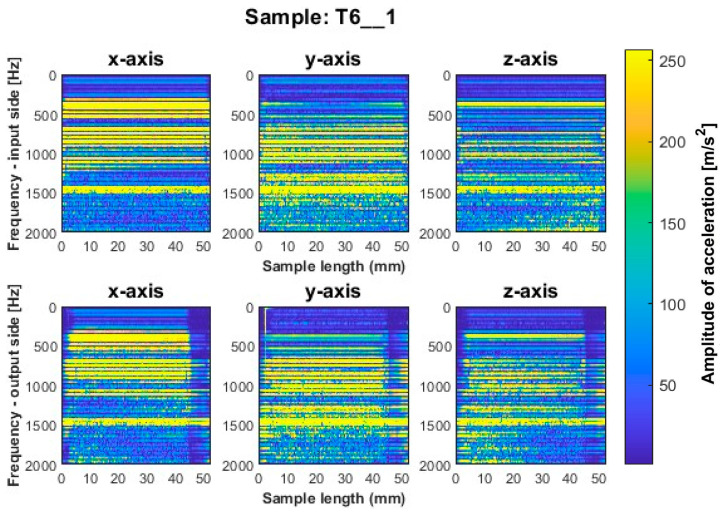
Vibration spectrogram of the acceleration signal for sample T6_1.

**Figure 12 sensors-23-06398-f012:**
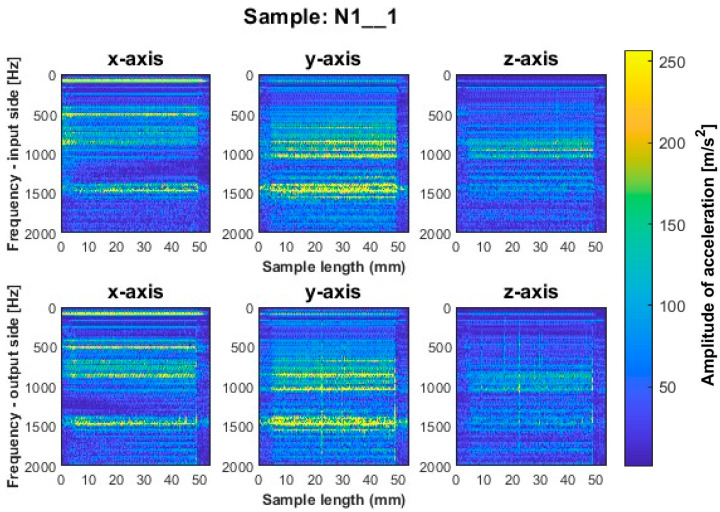
Vibration spectrogram of the acceleration signal for sample N1_1.

**Figure 13 sensors-23-06398-f013:**
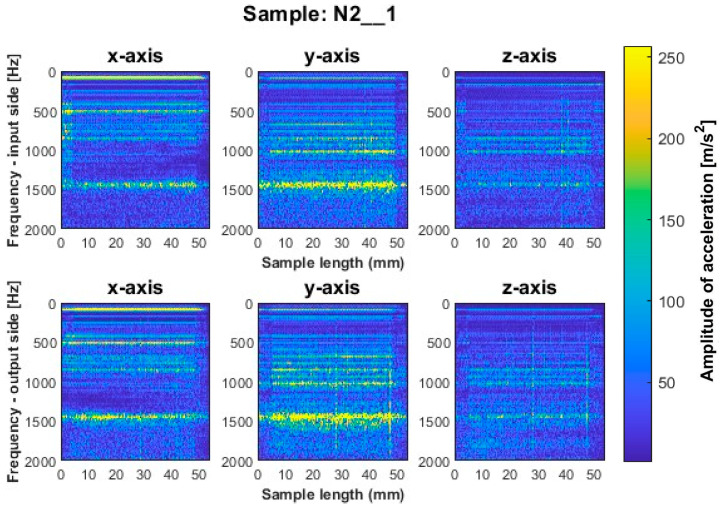
Vibration spectrogram of the acceleration signal for sample N2_1.

**Figure 14 sensors-23-06398-f014:**
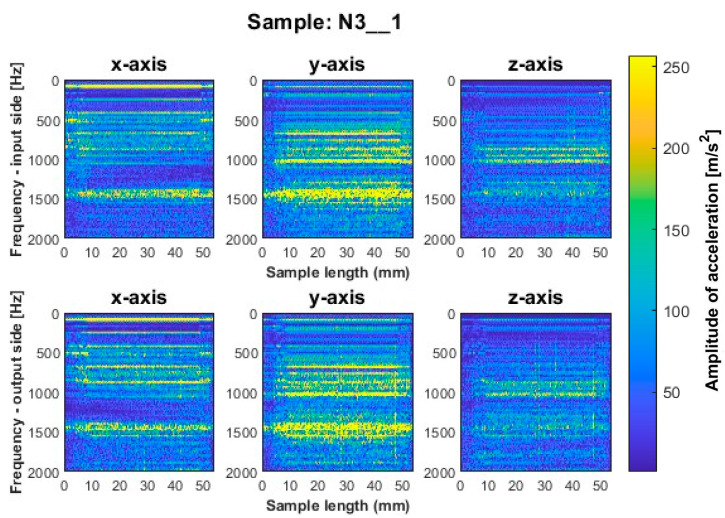
Vibration spectrogram of the acceleration signal for sample N3_1.

**Figure 15 sensors-23-06398-f015:**
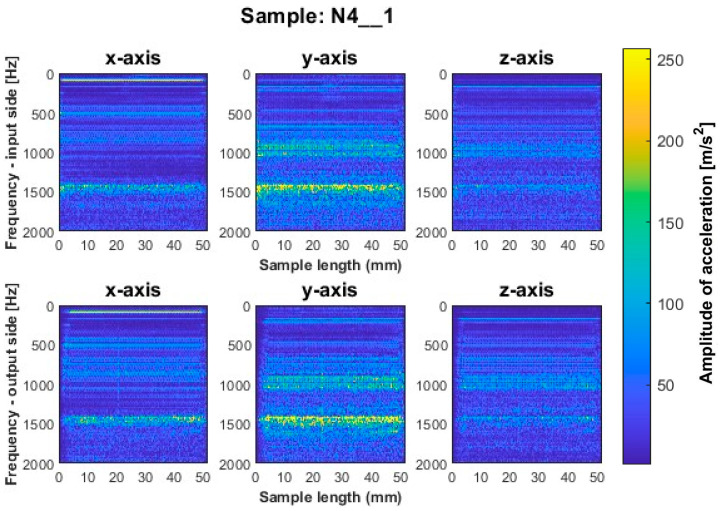
Vibration spectrogram of the acceleration signal for sample N4_1.

**Figure 16 sensors-23-06398-f016:**
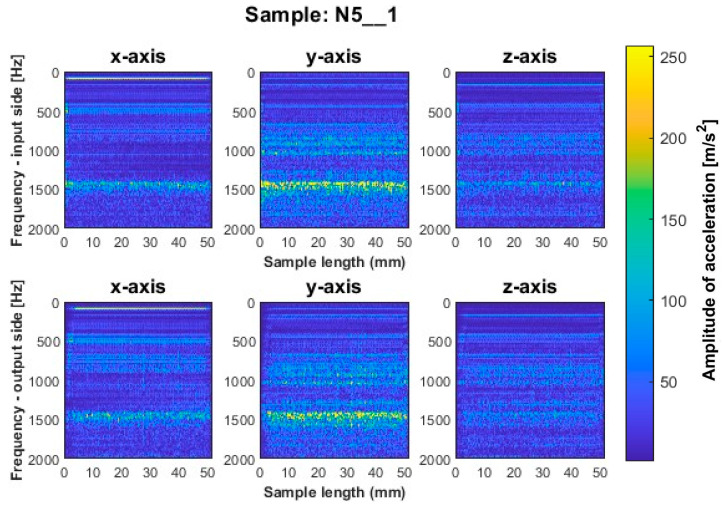
Vibration spectrogram of the acceleration signal for sample N5_1.

**Figure 17 sensors-23-06398-f017:**
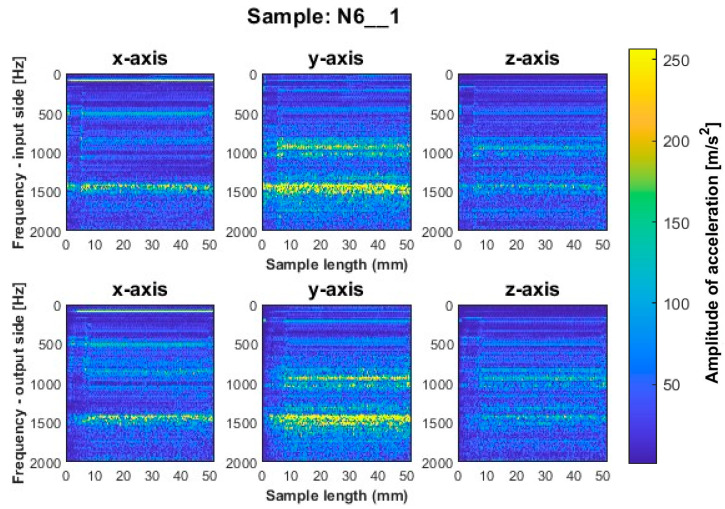
Vibration spectrogram of the acceleration signal for sample N6_1.

**Figure 18 sensors-23-06398-f018:**
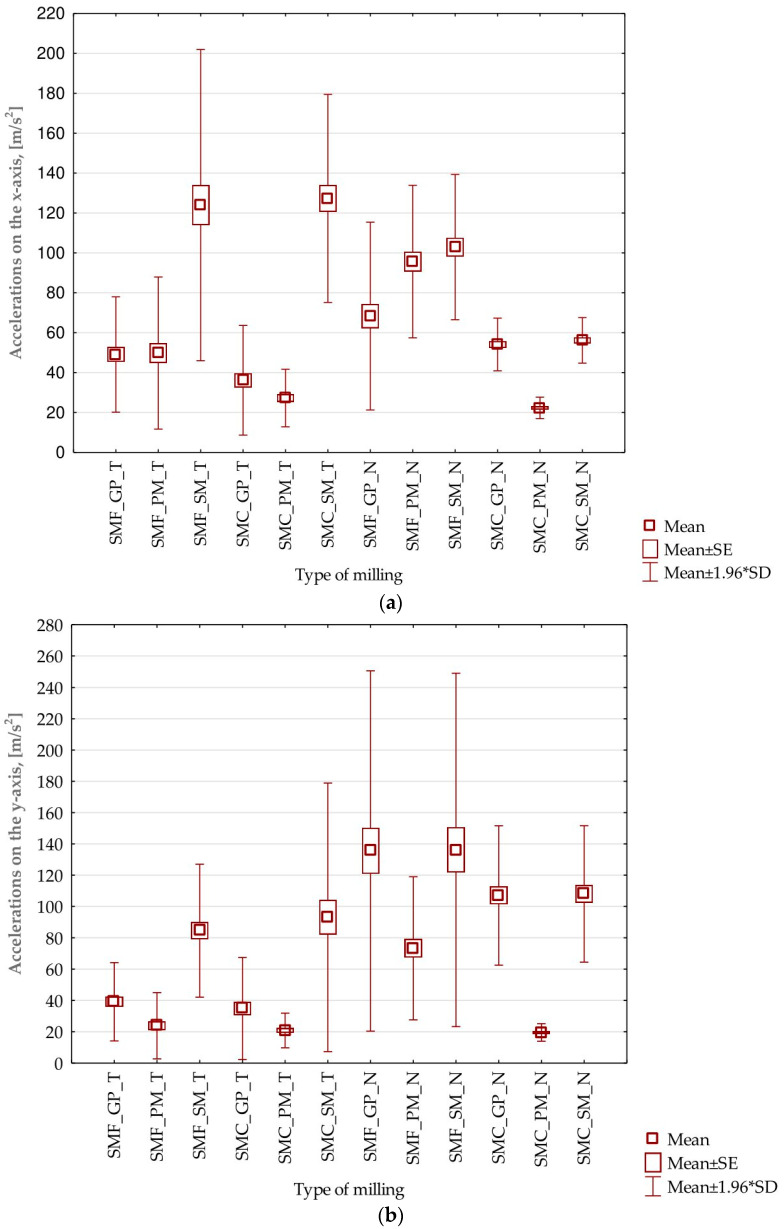
Acceleration values for different samples processing: (**a**) on the *x*-axis; (**b**) on the *y*-axis; (**c**) on the *z*-axis; SMF—side milling with increased engagement of the front part; SMC—side milling with increased involvement of the cylindrical part; GP—for general purpose; PM—for high-performance machining; SM—for high-speed machining; T—titanium alloy Ti6Al4V (grade 5); N—nickel alloy Inconel 625; SE—standard error; SD—standard deviation.

**Figure 19 sensors-23-06398-f019:**
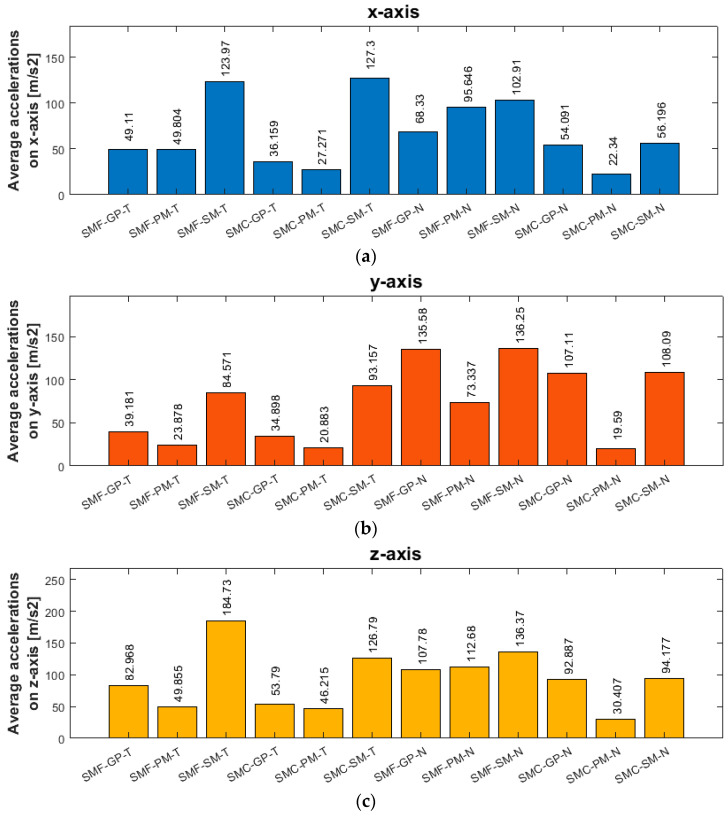
Average acceleration values for different samples processing: (**a**) on the *x*-axis; (**b**) on the *y*-axis; (**c**) on the *z*-axis; SMF—side milling with increased engagement of the front part; SMC—side milling with increased involvement of the cylindrical part; GP—for general purpose; PM—for performance machining; SM—for high-speed machining; T—titanium alloy Ti6Al4V (grade 5); N—nickel alloy Inconel 625.

**Table 1 sensors-23-06398-t001:** The chemical composition of titanium alloy Ti6Al4V—grade 5 [32].

Element	Ti	Al	V	Fe	C	N	O	H
Percentage [%]	balance	5.5–6.75	3.5–4.5	≤0.4	≤0.08	≤0.05	≤0.2	<0.015

**Table 2 sensors-23-06398-t002:** The mechanical properties of titanium alloy Ti6Al4V—grade 5 [32].

Mechanical Properties	Value	Unit
Tensile strength R_m_	min. 895	MPa
Yield strength 0.2%	min. 828	MPa
Elongation at break	min. 10	[%]
Density	4.52	g/cm^3^

**Table 3 sensors-23-06398-t003:** The chemical composition of nickel alloy Inconel 625 [33].

Element	Ni	Cr	Mo	Nb	Fe	C	Mn	Si	S	Al	Ti	P	Co
Percentage [%]	≥58	20–23	8–10	3.15–4.15	≤5	≤0.1	≤0.5	≤0.5	≤0.015	≤4.4	≤0.4	≤0.015	≤1

**Table 4 sensors-23-06398-t004:** The mechanical properties of nickel alloy Inconel 625 [33].

Mechanical Properties	Value	Unit
Tensile strength R_m_	min. 760	MPa
Yield strength 0.2%	min. 380	MPa
Elongation at break	min. 35	[%]
Density	8.44	g/cm^3^

**Table 5 sensors-23-06398-t005:** Type and description of tools used in the experiment.

Tool Number	Tool Model	Tool Description
Tool 1	JS554100E2R050.0Z4-SIRA	For general purpose
Tool 2	JS754100E2C.0Z4A-HXT	For high-performance machining
Tool 3	JH730100D2R100.0Z7-HXT	For high-speed machining

**Table 6 sensors-23-06398-t006:** Specification of the geometry of the tool we have used (own elaboration, based on [34,35,36]); chamfer is a symmetrical sloping surface at an edge or corner.

Parameter	Tool 1	Tool 2	Tool 3
Diameter	10 mm	10 mm	10 mm
Maximum depth of cut	22 mm	20 mm	31 mm
Overall length	72 mm	72 mm	72 mm
Number of cutting edges	4	4	7
Coating	SIRON-A	HXT	HXT
Flute helix angle	48°	48°	34°
Lead angle	0°	0°	0°
Corner chamfer width/corner round radius	0.5 mm (round)	0.125 mm (chamfer)	1 mm (round)

**Table 7 sensors-23-06398-t007:** Cutting parameters were adopted during the experiment.

Material	Sample	Tool	Cutting Speed V_c_ [m/min]	Feed Rate V_f_ [mm/min]	Depth of Cut a_p_ [mm]	Radial Depth a_e_ [mm]
Titanium alloy Ti6Al4V (grade 5)	T1_1	Tool 1	100	255	2 (8 passes)	4 (1 pass)
	T2_1	Tool 2	100	255	2 (8 passes)	4 (1 pass)
	T3_1	Tool 3	100	255	2 (8 passes)	4 (1 pass)
	T4_1	Tool 1	100	255	16 (1 pass)	0.5 (8 passes)
	T5_1	Tool 2	100	255	16 (1 pass)	0.5 (8 passes)
	T6_1	Tool 3	100	255	16 (1 pass)	0.5 (8 passes)
Nickel alloy Inconel 625	N1_1	Tool 1	40	255	2 (8 passes)	4 (1 pass)
	N2_1	Tool 2	40	255	2 (8 passes)	4 (1 pass)
	N3_1	Tool 3	40	255	2 (8 passes)	4 (1 pass)
	N4_1	Tool 1	40	255	16 (1 pass)	0.5 (8 passes)
	N5_1	Tool 2	40	255	16 (1 pass)	0.5 (8 passes)
	N6_1	Tool 3	40	255	16 (1 pass)	0.5 (8 passes)

**Table 8 sensors-23-06398-t008:** Calibration data of PCB-356B08 vibration sensor (based on manufacturer’s documentation).

Parameter	*x*-axis	*y*-axis	*z*-axis
Voltage sensitivity [mV/g]	97.4	100.2	97
Frequency range [Hz]	0.5–5000	0.5–5000	0.5–5000
Output voltage [V]	10.6	10.6	10.8

**Table 9 sensors-23-06398-t009:** Summary of statistical analysis of results on the *x*-axis.

Type of Milling	Valid N	Mean	Median	Minimum	Maximum	Variance	Standard Deviation	Standard Error
SMF_GP_T	16	49.11	46.46	28.10	73.23	217.41	14.74	3.69
SMF_PM_T	16	49.80	48.98	23.12	95.75	378.16	19.45	4.86
SMF_SM_T	16	123.97	116.30	78.92	242.30	1583.50	39.79	9.95
SMC_GP_T	16	36.16	29.97	25.58	78.24	196.36	14.01	3.50
SMC_PM_T	16	27.27	26.57	18.33	43.14	54.28	7.37	1.84
SMC_SM_T	16	127.30	125.70	81.59	167.81	707.31	26.60	6.65
SMF_GP_N	16	68.33	62.10	40.26	137.84	576.39	24.01	6.00
SMF_PM_N	16	95.65	96.46	51.02	127.27	380.52	19.51	4.88
SMF_SM_N	16	102.91	105.30	60.82	136.20	344.62	18.56	4.64
SMC_GP_N	16	54.09	55.06	41.96	63.00	45.47	6.74	1.69
SMC_PM_N	16	22.34	21.22	18.99	27.39	7.49	2.74	0.68
SMC_SM_N	16	56.20	58.10	45.90	66.60	34.09	5.84	1.46

**Table 10 sensors-23-06398-t010:** Summary of statistical analysis of results on the *y*-axis.

Type of Milling	Valid N	Mean	Median	Minimum	Maximum	Variance	Standard Deviation	Standard Error
SMF_GP_T	16	39.18	38.38	15.29	58.69	162.57	12.75	3.19
SMF_PM_T	16	23.88	20.37	12.01	53.74	116.72	10.80	2.70
SMF_SM_T	16	84.57	80.26	59.15	125.85	468.98	21.66	5.41
SMC_GP_T	16	34.90	28.63	23.30	78.95	276.91	16.64	4.16
SMC_PM_T	16	20.88	19.21	15.80	34.40	31.90	5.65	1.41
SMC_SM_T	16	93.16	81.03	62.00	246.42	1914.99	43.76	10.94
SMF_GP_N	16	135.58	130.47	71.49	266.57	3451.39	58.75	14.69
SMF_PM_N	16	73.34	66.42	41.53	130.17	542.59	23.29	5.82
SMF_SM_N	16	136.25	135.40	73.20	264.13	3319.93	57.62	14.40
SMC_GP_N	16	107.11	95.85	80.80	143.43	516.12	22.72	5.68
SMC_PM_N	16	19.59	19.41	15.79	26.09	8.18	2.86	0.72
SMC_SM_N	16	108.09	99.28	81.60	141.63	495.24	22.25	5.56

**Table 11 sensors-23-06398-t011:** Summary of statistical analysis of results on the *z*-axis.

Type of Milling	Valid N	Mean	Median	Minimum	Maximum	Variance	Standard Deviation	Standard Error
SMF_GP_T	16	82.97	87.94	43.54	111.64	408.22	20.20	5.05
SMF_PM_T	16	49.85	45.66	24.20	97.74	372.38	19.30	4.82
SMF_SM_T	16	184.73	180.83	86.74	328.66	3780.15	61.48	15.37
SMC_GP_T	16	53.79	48.24	36.61	88.70	275.22	16.59	4.15
SMC_PM_T	16	46.21	42.30	24.84	80.44	336.17	18.34	4.58
SMC_SM_T	16	126.79	122.70	99.25	159.39	322.83	17.97	4.49
SMF_GP_N	16	107.78	89.63	57.83	245.41	2562.68	50.62	12.66
SMF_PM_N	16	112.68	103.23	73.35	208.45	1116.26	33.41	8.35
SMF_SM_N	16	136.37	116.36	94.51	250.30	2214.03	47.05	11.76
SMC_GP_N	16	92.89	95.72	43.37	126.63	554.98	23.56	5.89
SMC_PM_N	16	30.41	30.46	25.03	36.48	9.84	3.14	0.78
SMC_SM_N	16	94.18	99.80	44.33	128.61	632.85	25.16	6.29

## Data Availability

All data are available in the manuscript.

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
