# Peer review of "Evaluation of the Vibration Signal during Milling Vertical Thin-Walled Structures from Aerospace Materials"

_sensors, 2023, doi:10.3390/s23146398_

Round 1

Reviewer 1 Report

The authors presented a study on analysis of vibration during milling two types of alloys by evaluation of their time-frequency responses using STFT. The goal of the study is unclear and the presented results are not convincing. The manuscript requires deep revision before further consideration.

  1. The abstract seems to be too long. Its shortening is recommended by its limiting to the most necessary information.
  2. The Introduction contains a lot of very fundamental information, which is widely known, therefore, it should be shortened. The authors discussed classical methods, like DFT and STFT, however, both transforms have significant deficiencies. Recently, multiscale transforms are usually used for analysis of this kind of non-stationary signals, e.g. wavelet transforms. The authors should consider current applications of transforms for signal processing and provide a justification for using STFT in this study.
  3. The Introduction does not cover a review of similar studies, it is also not clear which specific features are of the interest and what is the purpose of using of STFT instead of DFT. Finally, the original input of the authors within this study is unclear, which is connected to the previously mentioned questions. This part needs to be revised.
  4. Line 193: which oils were mixed?
  5. Please provide more details on STFT analysis: frequency range, justification for the selected lengths of window and Hanning window, which types of components are expected in the spectrograms?
  6. The performed qualitative analysis in section 3 does not bring useful information from the point of view of planning the machining process and the purposefulness of these results is unclear. A deep quantitative analysis should be performed to demonstrate the specific features and differences characteristic for the process. The observations need to be closely related with the phenomena occurring during the cutting process.
  7. To draw conclusions from the study, it is necessary to ensure statistical repeatability of the results. Please address to the number of performed trials and statistical evaluation.
  8. The conclusions need to be revised by structuring the results, adding quantitative measures that will represent particular cases, revise the terminology (e.g. using “chaotic” is not appropriate in this context).

Check lines 42, 126, 250, 263-264

Author Response

Dear reviewer,

Thank you for the feedback you provided and the list of possible improvements. Specific responses to the issues you have raised:

  1. The goal of the study is unclear and the presented results are not convincing.

The objective has been defined in the main body of the article and it is also contained in the title of the paper.

  1. The abstract seems to be too long. Its shortening is recommended by its limiting to the most necessary information.

We agree, the abstract has been shortened.

  1. The Introduction contains a lot of very fundamental information, which is widely known, therefore, it should be shortened. The authors discussed classical methods, like DFT and STFT, however, both transforms have significant deficiencies. Recently, multiscale transforms are usually used for analysis of this kind of non-stationary signals, e.g. wavelet transforms. The authors should consider current applications of transforms for signal processing and provide a justification for using STFT in this study.

In our opinion - not everybody interested in using aerospace materials is necessarily an expert in signal processing. We use a simple method to determine the basic parameters. The methodology and conduct of the experiment as well as the conditions of the study are included.

  1. The Introduction does not cover a review of similar studies, it is also not clear which specific features are of the interest and what is the purpose of using of STFT instead of DFT. Finally, the original input of the authors within this study is unclear, which is connected to the previously mentioned questions. This part needs to be revised.

The relation between DFT and STFT is now explained. There is lack of publications discussing STFT during milling vertical thin-wall components; we provide a short introduction for this reason. We refer to processing work to show the applicability of STFT in this area. In our opinion, the use of STFT in the machining of thin-walled structures is important because of the possibility of observing changes in time/sample length, hence it correlates well with other indicators we discuss. As stated in the conclusions, a comparison with surface parameters deserves to be made.  Such comparisons have already been made and will be published in the future.

  1. Line 193: which oils were mixed?.

Changed to: “A mixture of 15 vol. % oil and 85 vol. % water was used when machining the samples”.

  1. Please provide more details on STFT analysis: frequency range, justification for the selected lengths of window and Hanning window, which types of components are expected in the spectrograms?

The frequency range is provided in the graphs, more information about certain parameters is now included. As now stated, the built-in spectrogram function in MatLAB with the Hanning window was used.

  1. The performed qualitative analysis in section 3 does not bring useful information from the point of view of planning the machining process and the purposefulness of these results is unclear. A deep quantitative analysis should be performed to demonstrate the specific features and differences characteristic for the process. The observations need to be closely related with the phenomena occurring during the cutting process.

“Deep quantitative analysis” is undefined, but statistical analysis was introduced. We have 18 figures, 9 tables and provide a number of conclusions.

  1. To draw conclusions from the study, it is necessary to ensure statistical repeatability of the results. Please address to the number of performed trials and statistical evaluation.

A statistical analysis is included in the new version.

  1. The conclusions need to be revised by structuring the results, adding quantitative measures that will represent particular cases, revise the terminology (e.g. using “chaotic” is not appropriate in this context).

The section with conclusions has been reworded.

I hope that the changes correspond to the comments sent and the article was prepared as expected.

Best Regards,

Authors

Reviewer 2 Report

The research content of this article is the vibration characteristics of cutting specimens using different cutting tools at different cutting depths and radial depths. Fig. 5 to Fig. 16 are the core results of this study. The reviewer has some issues with the content in the figure, please explain.

1. The vibration characteristic is a time history, and the abscissa of these graphs is sample length. How is the time related to the sample length?

2. According to the experiments design in Table 7, the milling of each surface of the sample requires 8 passes. Are these spectrograms the average of 8 processes or one process?

3. Each figure from Fig. 5 to Fig. 16 has 2 rows and 6 spectrograms. Under what experimental conditions were the input side above and output side below measured? There are significant difference between the upper and lower rows in the figures. What are the specific reason for these difference? No analysis explanation are provided in the article.

4. The frequency range of the accelerometer used in these experiments is 0.5-5000Hz, but the upper frequency limit shown in Figures 5 to 16 has reached 12000Hz, which exceeds the analysis frequency range. It is recommended to change the displayed frequency in the figure to 0-2000Hz.

5. What information does the color in the spectrogram represent? Vibration amplitude? And the unit is missing. Is it m/s2 ,dB or others?

6. There is a lack of analysis and discussion of image response information in the summary and conclusion.

Author Response

Dear reviewer,

     We appreciate the feedback you sent and the list of possible improvements. Specific answers:

  1. The vibration characteristic is a time history, and the abscissa of these graphs is sample length. How is the time related to the sample length?

The relation was determined based on Eq. (3).

  1. According to the experiments design in Table 7, the milling of each surface of the sample requires 8 passes. Are these spectrograms the average of 8 processes or one process?

As mentioned - spectrograms were made for the last pass for the input and output sides (see Fig. 2.). In the revised version of the article, a statistical analysis has been introduced, which includes amplitude values from all 16 passes.

  1. Each figure from Fig. 5 to Fig. 16 has 2 rows and 6 spectrograms. Under what experimental conditions were the input side above and output side below measured? There are significant difference between the upper and lower rows in the figures. What are the specific reason for these difference? No analysis explanation are provided in the article.

Each sample contained an input and output side; see Fig. 2. The machining conditions for each sample are provided in Table 7. The differences are due to the fact that the input side had a thin wall somewhat increased by the thickness of the cut layer (this is the penultimate pass and last on this side) while the output side is the surface obtained during the last pass.

  1. The frequency range of the accelerometer used in these experiments is 0.5-5000Hz, but the upper frequency limit shown in Figures 5 to 16 has reached 12000Hz, which exceeds the analysis frequency range. It is recommended to change the displayed frequency in the figure to 0-2000Hz.

We agree. Above 2000 Hz no significant results were observed, hence the spectrograms were corrected as recommended.

  1. What information does the color in the spectrogram represent? Vibration amplitude? And the unit is missing. Is it m/s2 ,dB or others?

A label has been added; acceleration was measured during experiment, the unit is m/s2).

  1. There is a lack of analysis and discussion of image response information in the summary and conclusion.

The results analysis and conclusions have been modified.

I hope that the changes correspond to the comments sent and the article was prepared as expected.

Best Regards,

Authors

Reviewer 3 Report

Attached

Author Response

Dear reviewer,

        we appreciate the feedback you have provided and the list of improvements. The following changes have been made:

  1. The parameters taken for both samples are same. However, the cutting speed is different. Have the authors considered the cutting speed of 40 m/min incase of Ti-alloy?

The aim of the study was to evaluate the process during milling thin wall samples with an increased section of the cut layer, which are connected with material removal rate. MMR depends on the feed rate, the depth of the cut and the radial depth. The value of material removal rate is not dependent on the cutting speed. As mentioned in lines 160-163, the parameters and conditions were selected according to the recommendations of the cutting tool manufacturer. Nickel alloy is a harder material, hence lower cutting speeds should be used than for the titanium alloy. The cutting speed was selected as the middle value of the tool manufacturer's recommended range. The minimum cutting speed of titanium alloy with certain tools is about 60 m/min. In our opinion a middle value of cutting speed is appropriate for a comparison a results.

  1. Line: 193, A mixture of 15% oil and 85% oil was used when machining the samples. However, water-oil emulsion was used.

Changed to: “A mixture of 15 vol. % oil and 85 vol. % water was used when machining the samples”.

  1. The discussion part needs to be supported by proper citation.

The results analysis and conclusions have been amplified and more literature is cited, including now four publications in Sensors.

I hope that the changes correspond to the comments sent and the article was prepared as expected.

Best Regards,

Authors

Round 2

Reviewer 1 Report

The authors performed necessary corrections of the manuscript together with the provided answers to the comments from the first round of review. Although quantitative analysis was not performed according to the suggestion, the performed statistical analysis partially fulfill the mentioned comment, i.e. the results with statistics are not case-specific anymore and have practical applicability. I recommend this manuscript for a publication in its present form.

Some sentences need rephrasing to improve their grammar correctness.

Reviewer 2 Report

The paper can be accepted.